# The Impact of COVID-19 on High School Student-Athlete Experiences with Physical Activity, Mental Health, and Social Connection

**DOI:** 10.3390/ijerph18073515

**Published:** 2021-03-29

**Authors:** Heather A. Shepherd, Taffin Evans, Srijal Gupta, Meghan H. McDonough, Patricia Doyle-Baker, Kathy L. Belton, Shazya Karmali, Samantha Pawer, Gabrielle Hadly, Ian Pike, Stephanie A. Adams, Shelina Babul, Keith Owen Yeates, Daniel C. Kopala-Sibley, Kathryn J. Schneider, Stephanie Cowle, Pamela Fuselli, Carolyn A. Emery, Amanda M. Black

**Affiliations:** 1Sport Injury Prevention Research Centre, Faculty of Kinesiology, University of Calgary, Calgary, AB T2N 1N4, Canada; taffin.evans@ucalgary.ca (T.E.); srijal.gupta1@ucalgary.ca (S.G.); kjschnei@ucalgary.ca (K.J.S.); caemery@ucalgary.ca (C.A.E.); ablack@ucalgary.ca (A.M.B.); 2Alberta Children’s Hospital Research Institute, University of Calgary, Calgary, AB T2N 1N4, Canada; pdoyleba@ucalgary.ca (P.D.-B.); kyeates@ucalgary.ca (K.O.Y.); daniel.kopalasibley@ucalgary.ca (D.C.K.-S.); 3Hotchkiss Brain Institute, University of Calgary, Calgary, AB T2N 4N1, Canada; 4O’Brien Institute for Public Health, University of Calgary, Calgary, AB T2N 4Z6, Canada; 5Faculty of Kinesiology, University of Calgary, Calgary, AB T2N 1N4, Canada; meghan.mcdonough@ucalgary.ca; 6School of Architecture, Planning and Design, University of Calgary, Calgary, AB T2N 1N4, Canada; 7Injury Prevention Centre, School of Public Health, University of Alberta, Edmonton, AB T6G 2E1, Canada; kbelton@ualberta.ca; 8BC Injury Research and Prevention Unit, BC Children’s Hospital, Vancouver, BC V6H 3V4, Canada; shazya.karmali@bcchr.ca (S.K.); samantha.pawer@bcchr.ca (S.P.); gabrielle.hadly@alumni.ubc.ca (G.H.); ipike@bcchr.ca (I.P.); sbabul@bcchr.ca (S.B.); 9Department of Pediatrics, University of British Columbia, Vancouver, BC V6T 1Z4, Canada; 10Werklund School of Education, University of Calgary, Calgary, AB T2N 1N4, Canada; stephanie.adams1@ucalgary.ca; 11Clinical Psychology, School of Health in Social Science, University of Edinburgh, Edinburgh EH8 9AG, UK; 12Institute for Sport, Physical Education and Health Sciences, Moray House School of Education and Sport, University of Edinburgh, Edinburgh EH8 8QA, UK; 13Department of Psychology, University of Calgary, Calgary, AB T2N 1N4, Canada; 14Cumming School of Medicine, University of Calgary, Calgary, AB T2N 4N1, Canada; 15Department of Psychiatry, University of Calgary, Calgary, AB T2N 4N1, Canada; 16Mathison Centre for Mental Health Research and Education, University of Calgary, Calgary, AB T2N 4Z6, Canada; 17Sport Medicine Centre, Faculty of Kinesiology, University of Calgary, Calgary, AB T2N 1N4, Canada; 18Parachute, Toronto, ON M4P 1E8, Canada; scowle@parachute.ca (S.C.); pfuselli@parachute.ca (P.F.)

**Keywords:** physical activity, mental health, social connection, youth-athletes, COVID-19, pandemic

## Abstract

COVID-19 restrictions led to reduced levels of physical activity, increased screen usage, and declines in mental health in youth; however, in-depth understandings of the experiences of high school student-athletes have yet to be explored. To describe the experiences of the COVID-19 pandemic on student-athletes’ physical activity, social connection, and mental health, 20 high school student-athletes living in Calgary, Alberta participated in semi-structured interviews, designed using phenomenography. Participants reported variations in physical activity, social connections, and mental health which were influenced by stay-at-home restrictions and weather. Access to resources, changes to routines, online classes, and social support all influenced engagement in physical activity. School and sports provided opportunities for in-person social connections, impacted by the onset of the pandemic. Participants reported their mental health was influenced by social connections, online classes, and physical activity. Findings from this study will inform the development of resources for high school student-athletes amidst COVID-19.

## 1. Introduction

In 2020, the novel coronavirus disease, COVID-19, spread rapidly around the globe [1]. To control the spread of COVID-19, health authorities enacted policies to increase physical distancing and limit person-to-person contact [2,3,4]. This included closing schools, community centres, parks, athletic and fitness facilities, and halting organized sports [5,6]. Canadian health authorities provided guidelines to maintain physical activity safely, including engaging with people from the same household or small groups, while maintaining physical distancing [7,8]. Options for physical activity included in-home exercise programs (e.g., yoga, online fitness classes), outside activity during non-peak hours, and appropriate hand hygiene practices before and after activity [7,8,9,10,11].

COVID-19 has led to the rapid dissemination of research examining the impact of the pandemic on physical activity, social connections, and mental health. A study of parents in Italy and Spain reported that >85% of their children decreased physical activity levels, increased screen time, and increased maladaptive emotional and behavioural signs [12]. In Germany, increased isolation and decreased social connections were associated with increased psychological distress [13]. In China, 40% of youth experienced psychological distress (e.g., post-traumatic stress disorder [14%], negative coping, stress) after COVID-19 was announced as a public health emergency [14].

In the United States, a study of high school student-athletes reported elevated self-reported anxious and depressive symptoms in females when compared to males [15]. McGuine et al. [15] also reported increased self-reported anxious and depressive symptomatology in grade 12 students, when compared to lower grades. Schools may provide opportunities for mental health supports (e.g., guidance counsellor) [16], and without these supports in place, coupled with increased isolation, change in routine, and decreased opportunities for socialization, the mental health in high school students is of concern.

Physical activity and social connection are important determinants of mental health [17,18,19,20]. Researchers reported that youth who participate in team sports had better mental health scores than youth who engaged in individual sports [18]. This is in contrast to the results reported by McGuine et al. [15] who found that during the first few months of COVID-19, team sport athletes self-reported more anxious and depressive symptoms compared to individual sport athletes. As such, the effect of COVID-19 may be more difficult for team sport athletes.

In addition, physical activity of any kind has been positively associated with mental health among youth [20,21,22]. Schools traditionally facilitate opportunities for group-based physical activity and social connection through physical education and extra-curricular sport activities [23]. However, physical distancing measures enacted during COVID-19 and the suspension of in-person schooling might reduce important social connections and physical activity opportunities for student-athletes.

The impact of the COVID-19 pandemic on Canadian youth is not yet well understood. Few researchers describe the experiences and perceptions of Canadian youth during the pandemic. Existing studies rely on survey methods to examine youth experiences during COVID-19; however, they are limited in understanding the depth and variations of youth perceptions during COVID-19 and have not focused on the Canadian student-athlete population.

Stopping or modifying school, sports, and/or recreational opportunities may greatly impact health behaviours including engagement in physical activity, social connections, and mental health among high school student-athletes. The aim of this study was to describe and interpret Canadian high school student-athletes’ experiences with physical activity, mental health, and social connections during the COVID-19 pandemic.

## 2. Methods

### 2.1. Methodological Paradigm

This study employed a second-order phenomenological perspective (phenomenography) [24]. Phenomenography seeks to understand variation in the subjective nature of experiences, perceptions, and conceptualizations of a phenomenon [24,25]. It aims to characterize descriptive categories that differ from one another, and can be used to identify health-related concepts that can be targeted for change [26].

### 2.2. Participants

High school student-athletes from Calgary, Alberta were purposively sampled from participants currently enrolled in the three-year pan-Canadian Surveillance in High Schools to Reduce Concussions (SHRed Concussions) cohort study, and who engaged in at least one of 10 high-risk concussion sports (e.g., rugby, ice hockey, soccer). Participants were initially invited to participate in this sub-study by a member of the research team and were provided an overview of the study purpose. A total of 166 participants were contacted, 99 of whom expressed interest in participation. The 99 participants were then sampled for variation on: (1) sex; (2) gender; and (3) sport participation (i.e., whether they participated in a spring season sport or not). Participants were then purposively sampled, using maximum variation sampling, including variation in sport participation, season of sport, and individual or team sport participation (see Figure 1). Participants were then invited to participate in an interview. Sport information was retrieved from a physical activity participation survey included as part of the larger study.

Participants were informed that all information collected would remain confidential and de-identified. Participants provided verbal consent to participate. The study was approved by the University of Calgary Conjoint Health Research Ethics Board (#REB18-2107).

### 2.3. Data Collection

Semi-structured interviews were conducted via phone or Zoom and were scheduled to be 45–60 min in duration (see Appendix A for interview guide). At the conclusion of each interview, each participant received a debrief e-mail outlining mental health resources and support services. All interviews were completed by HAS and TE (first and second author, respectively) in June 2020, during Phase 1 when certain COVID-19 restrictions (e.g., day camps, summer school, parks) were lifted in Calgary, Alberta. Interviews were recorded and transcribed verbatim via Rev.com [27]. Each transcript was validated against the audio recordings by two reviewers.

### 2.4. Data Analysis

The data analysis process was guided by steps outlined by Kinnunen and Simon [25]: (1) data familiarization by reading transcripts and listening to all audio recordings of interviews; (2) identifying emergent codes; (3) refining and redefining categories; (4) finalizing categories and providing a thorough description of the categories; and (5) discussing relationships between categories. Sex/gender and sport participation were considered during Step 5. Although listed stepwise, the process was iterative. The lead author completed all steps in the analysis in consultation with the research team. NViVO 12 [28] was used to support the data analysis process.

#### Bracketing and Reflexivity

To enhance trustworthiness, bracketing and reflexivity practices were adhered to throughout the research process [29]. Bracketing refers to researchers suspending judgement and focusing on the inductive analysis of experiences as described by the participants [29]. Prior to data collection, all members of the data collection and analysis team underwent training on bracketing and reflexivity. Interviewers and transcript reviewers kept reflexive memos to record their interpretations, their experiences of the interview, and potential ways they may have influenced interviews (e.g., interpretation of participant responses, rapport building) or interpreted transcriptions. To increase credibility, peer debriefing was completed after interviews and after each stage of the data analysis process. To support dependability, an audit trail was used to track all aspects of decision making. Additional author information is provided in Appendix A.

## 3. Results

Thirty-six participants were contacted to participate, of which 20 completed an interview. Fourteen participants could not be reached, and two participants declined to participate (see Figure 1). All interviews were conducted between 2 and 9 June 2020 and ranged from 28 to 80 min in length. Interviews were conducted via telephone (*n* = 19) or Zoom (*n* = 1), based on participant preference. All participants (10 males and 10 females, ages 15–17 years) identified as cis-gendered. Five female and five male participants typically participated in a high school spring sport (March–June 2020) and five female and five male participants typically participated in a high school sport in another season (see Table 1). Although sampled on sex, gender, and spring sport participation, no reported differences on experiences of physical activity, mental health, nor social connections were apparent between males and females or based on season of sport participation.

### 3.1. Overview

The onset of the COVID-19 pandemic in Calgary, Alberta, when school and organized sports were cancelled (15 March 2020), led to variations in physical activity, mental health, and social connections amongst high school student-athletes (Figure 2). Participants reported that school and sports both provided opportunities for physical activity and social connections, and that these were altered with the onset of the pandemic. Poor weather, including snow and ice, affected student-athletes’ engagement in physical activity, predominantly limiting them to indoor exercises, disengaged from their peers, and reports of declines in mental health. Participant 2-M shared that the restrictions made him feel like he was, “*stuck inside*”, a sentiment shared by others. The lifting of restrictions (14–25 May 2020) coincided with an improvement in weather, increased opportunities for physical activity outdoors (e.g., biking, walking), and increased opportunities to connect with friends outside while observing physical distancing measures. Participants expressed how their increased physical activity and social connections were associated with an improvement in their mental health, *“I’m seeing my friends a little more…I’m walking around more…I’m not stuck at home” (P19-F)*, and that these improvements were a shift towards normal, or a “new normal”. Participant 10-M described his perspective of the lifting of restrictions, *“it’s better… you can kind of go outside and do stuff a little more…Like I’m feeling good you know. Everything’s kind of getting back to normal”.*

### 3.2. Variations in Mental Health and Well-Being

#### 3.2.1. Initial COVID-19 Restrictions Led to Increased Anxiety for Most Student-Athletes

Most participants reported their mental health worsened at the onset of COVID-19 restrictions, expressing feelings of anxiousness, fear, and shock. Participant 15-F shared, *“I had anxiety attacks all throughout the first week”.* Two participants reported no change in their mental health, and one reported the cancellation of school *“felt like…a longer and extended weekend” (P6-M)*.

#### 3.2.2. The Impact of Changes to Physical Activity and Mental Health during the Pandemic

Participants reported that prior to the COVID-19 pandemic, physical activity helped them to manage emotions, decreased their stress, and supported their physical and mental wellness. Physical activity and mental health declined at the onset of restrictions and at the onset of school resuming online for most participants, with many reporting improvements in physical and mental health at the time of the interview. Participant 20-M explained how the onset of restrictions “*affected me mentally ‘cause I was used to putting all my energy into a sport, but then suddenly I didn’t have anywhere to put that energy… it just made me feel very um, impatient or like, um, restless”*. A few participants shared that the COVID-19 restrictions enabled them to stay physically active throughout the pandemic, and that this helped maintain their mental health. Participant 5-M shared his experience saying he, *“just focus[es] on my weight training, just help alleviate stress”*.

### 3.3. The Suspension and Modification of School Routine Affected Social Connections, Physical Activity, and Mental Health

#### 3.3.1. The Loss of School and Sports Led to a Decrease in Social Connections

The cessation of school and sports decreased opportunities for social connection for some participants and led them to feel disconnected and lonely. Participant 1-F discussed how it affected her, stating, *“It impacted me a lot… how I stay connected with people is going to practices and talking to them there”.* This contrasts with a few participants who shared that the pandemic did not alter social connections, *“I only ever really see them in practice…and never really hung out with them, so we aren’t too close” (P18-F).*

#### 3.3.2. Online Physical Education Classes Lacked Physical Activity

Students enrolled in physical education classes discussed how their class, once focused on experiential fitness, activity, and health, transitioned to writing about how to engage in these activities, or to tracking their physical activity using the honour system. Participant 2-M shared, *“I found it was pointless because like we don’t really do any physical… We just like list down stuff that we did”*. One participant shared that her friend who attended a different school used a freely accessible phone application: *“it pretty much tracked their steps and their running” (P9-F)*, which she was interested in trialing for her physical education class.

#### 3.3.3. Returning to School Online Affected Mental Health and Well-Being Differently in Student-Athletes.

When school resumed online, many participants reported increased stress and frustration, citing lack of support from teachers, unclear expectations, and challenges with self-directed learning. Participant 9-F expressed that online *“classes stressed me out”*. However, others reported an improvement in their mental well-being when school resumed, citing the return to a routine helped structure their day. Participant 6-M shared how his prior experience with online learning helped ease the transition, “*I was doing online English at the time… so it was kind of just a transition into that”* and was less stressful than it may have been for his peers.

### 3.4. Changes to Physical Activity Programming Affected Engagement

#### 3.4.1. COVID-19 Restrictions Decreased the Amount and Intensity of Physical Activity

Participants reported their physical activity decreased at the onset of COVID-19 restrictions. Reasons for decreased activity included limited access to equipment, decreased motivation, and change to school and sport routines. Participant 4-F shared, *“I definitely stopped running as much. My endurance got a lot worse”* when describing her physical activity at the onset of restrictions. No participants explicitly reported that their physical activity levels increased at the onset of COVID-19 restrictions. However, most participants reported that their physical activity had increased at the time of the interview compared to the onset of restrictions, and some reported their activity had returned to pre-COVID-19 levels. Participant 15-F shared how her *“physical health is got, is definitely a lot better”*, at the time of interview than at the onset of the pandemic.

#### 3.4.2. Changes to Resources Led to Changes in Physical Activity

Access to a broad range of resources, including equipment, personnel, and knowledge of training plans, provided support to student-athletes to engage in physical activity pre-COVID-19. Access to resources was hampered by COVID-19 and affected student-athletes’ engagement in their pre-pandemic physical activity and sport-specific training. For some, the lack of resources required them to modify their activity, *“I wasn’t able to swim anymore, then I started…going on runs” (P11-F*), or work on other skills, *“I’ve been pretty bad with my cardio. And now it gives me a chance to work on that [sic]” (P2-M)*. Whereas other participants indicated that the lack of resources could be detrimental, *“I feel like you’re taking a step back from your potential” (P8-F).* Although lack of resources was a barrier to many, some participants were well-equipped to continue with their physical activity, *“we have a gym in the basement” (P13-M)*.

#### Creativity with Household Items as Fitness Equipment Increased Physical Activity

Some participants who lacked fitness equipment adapted household items either through their own creativity, or following the advice of their teachers, *“I have some PVC pipes that I used as my bar and then milk jugs on either side and used it as um, as like a bar weight” (P14-F)*.

#### Access to Training Programs Affected Engagement in Physical Activity and Social Connections

The cessation of school and organized sports resulted in the suspension of planned training (e.g., practices, physical education classes), resulting in some participants feeling at a loss of how to train, *“I didn’t have a coach to teach me anymore” (P13-M)*. Other student-athletes turned to friends, social media, or the Internet to educate themselves on how to develop a training plan, *“…so I Google like weight training routines” (P12-F)*. Meanwhile, some participants reported accessing training plans provided by their sports organizations, which provided structure for their training and guidance on what physical activities to perform. Participant 4-F discussed how her soccer team maintained a weekly schedule, *“Thursdays are always the running days and Saturdays are always the workout days”*, which facilitated her ongoing engagement in physical activity.

Some sports clubs and teams used social media platforms to complete training online via video conferencing which supported not only engagement in physical activity, but also supported connection with teammates and coaches. Access to virtual training was desired by participants who did not have access to such resources. Participant 8-F expressed her desire saying, *“since the rugby season got cancelled, I think it would have been cool if like we had like virtual practices or something”*.

### 3.5. Changes to Social Support and Connections In-Person and Online

#### 3.5.1. Social Support Was Important for Engaging in Physical Activity for Some Student-Athletes

Many participants shared that family, friend, and teammate support and encouragement (e.g., verbal praise) were more important to their engagement in physical activity during then pandemic than prior to the pandemic. For some, engaging in physical activity with their family members was helpful, *“we all just kind of connect in a way” (P11-F)*. Talking with friends or exercising together virtually helped some participants to follow through on their training, *“if we’re both doing it, then I’ll be more likely to actually follow through with it” (P11-F)*. However, other participants reported that they were self-motivated, *“I said to myself… maintain that goal” (P3-M)* and engaged in physical activity independent of their family, friends, teammates, or other external motivators (e.g., competition).

#### 3.5.2. Social Media and Physical Distancing Were Not Adequate Forms of Connection

Participants engaged in multiple means of connection, including texting, calling, social media, and, when permissible, physically distanced in-person meetings. Many participants shared that connecting with friends, teammates, and family was helpful to them, *“being able to be able to talk to other people has been—is really helpful” (P5-M)*. Social media allowed for communication over numerous platforms and platforms with video (e.g., Zoom, FaceTime) were preferred. Participant 18-F expressed her preference for interactive platforms saying, *“[I] feel like my personality is coming through… Whereas like texting… sometimes you don’t read it as well” (P18-F)*. However, some participants reported that the reliance on social media for communication was difficult and tiresome, *“[it] kinda gets like a little bit stale” (P16-M)*. Participants shared that although physical distancing was an improvement to social media, it still altered their normal interactions. Participant 17-M stated, *“it’s definitely different ‘cause you’re usually… be closer and… like, wrestle… you have to stay six feet apart and you can’t”* and Participant 4-F shared, that she *“miss[es] also being able to hug my friend”*. Some participants shared how the lack of in-person connection led to a decline in their mental health. Participant 7-M shared his experience stating, *“I’m not seeing those people that I have fun with for long amounts of time… the mind starts going into, into darker places, I guess, and isn’t feeling as happy”*.

## 4. Discussion

The aim of this study was to describe and interpret high school student-athletes’ experiences during COVID-19, and how their physical activity, mental health, and social connections were affected by the pandemic. Results highlight the multifaceted experiences and perspectives of these youth, with variations occurring both across and within individuals in all three domains (i.e., physical activity, mental health, social connections) during the initial three months of COVID-19 restrictions in Calgary, Alberta.

In Calgary, engagement in physical activity and in-person connections came to a halt for most high school-aged students when the Alberta Government announced, on March 15, 2020, that schools would be closed [30]. Schools traditionally provide opportunities for physical activity through physical education classes and extra-curricular sports, social relationships through classes and extra-curricular activities, and access to mental health resources and supports, such as school counsellors. Consistent with previous surveys [12,15,31,32,33], the stoppage or modifications to school and sports led to decreases in physical activity, social connections, and self-reported mental health for most student-athletes in our sample.

The descriptions provided by the participants highlight specific strategies that may assist with mitigating the negative effects of COVID-19 restrictions. Specifically, the loss of organized sports and loss of connection with teammates led student-athletes to express desire for virtual training provided by their coaches. Online options may increase access to regularly scheduled activity, create routine, and facilitate access to prescribed training plans, eliminating the need for adolescents to create their own. Virtual training may also increase social connection with teammates and coaches, especially if conducted in real-time, such as through Zoom or Skype. Training with others may also increase activity duration, intensity, or motivation as per the Köhler effect, which may further support engagement in physical activity [34,35]. The uptake of online platforms to support physical activity and social connection among student-athletes may be potential solutions for schools and for sport organizations. Student-athletes not presently enrolled in physical education class or extra-curricular sport may benefit from similar virtual training programs provided by community organizations.

The design of online training programs should include exercises that can be completed with limited or no equipment. This may help to reduce potential barriers, such as lower family socio-economic status, to participation in physical activity at home. Lower socio-economic standing is associated with decreased sport and physical activity participation and decreased access to resources to support physical activity [36,37,38,39]. Student-athletes who had access to resources, or maintained their physical activity throughout COVID-19, did not report changes to their overall health and wellness. This further supports evidence that physical activity acts a protective factor for mental health and physical health [11,40,41]. Thus, recommendations must consider accessibility for all student-athletes.

Efforts to raise awareness of freely accessible resources (e.g., Nike Run Club, ParticipACTION) by sports organizations, schools, and health organizations may increase physical activity and reduce barriers imposed by limited resources. Although research outlining types of at-home physical activity is limited, public health authorities continue to stress the importance of remaining physically active [42,43]. Further, resources outlining at-home physical activities have been developed, but further dissemination work is needed to translate this information to high school student-athletes and those stakeholders whom interact with them [10].

The change in school structure led to disruption in routines and subsequent declines in mental well-being among participants. The return to school online increased stress and frustration among many participants, who cited lack of preparation, poor communication with teachers, and unclear expectations as increased stressors. Available mental health resources should be promoted to student-athletes, including access to peer support, mentorship, and registered mental health professionals. Raising awareness of supports and resources that can be accessed through telehealth or virtual means should be highlighted considering the pandemic. Schools should also provide education to teachers on ways to support students via online learning. Implementation of existing mental health programs, such as MindMatters [44,45], a mental health promotion program, may help to support student-athlete mental health not only during the COVID-19 pandemic, but also after the pandemic has ended.

In-person communication provides opportunity for nonverbal communication to occur [46], and its absence was highlighted by the student-athletes when communicating via social media. A survey conducted in Britain found that 65% of students wanted to return to school for socialization purposes [47], highlighting the importance of school as a facilitator for developing and maintaining social relationships. In the wake of another wave of COVID-19 restrictions, schools should offer alternative means for students to connect, with increased opportunity for physically distanced, but in-person communications. If this is not possible, then opportunities for increased use of video communication, or social media platforms that permit two-way exchange, allowing for non-verbal communication, would be desirable.

Social media provided platforms for student-athletes to stay connected to family, friends, and teammates. Participants reported using different platforms for different forms of communication. For example, Snapchat and FaceTime were used to communicate with close friends, as they exhibit body language and tone of voice, whereas Pinterest and Instagram feeds were used to share resources (e.g., exercise routines). Social media can support mental health when it fosters social connections [48]; however, it can also decrease mental health, including feelings of increased isolation and depression among youth [33,48]. Although social media allowed for connections throughout COVID-19, the heavy reliance on social media may have negative health consequences for youth. Participants in this study shared that the reliance on social media became tiresome, and that reliance on it impacted their ability to communicate using non-verbal channels with their peers.

At the onset of COVID-19 in March 2020, Calgary was still experiencing winter weather and still in the midst of winter, most physical activity and socialization was restricted to indoors. The better weather and lifting of restrictions were associated with increased engagement in physical activity, particularly outside, increased social connections, with opportunities to connect with others via physical distance measures in outdoor spaces, and improvement in mental health. Most participants reported that their overall health and wellness improved at this time. Increased access to outdoor public spaces and green spaces may provide more opportunity for physical engagement and social connection, in addition to improvements in mental health [49,50]. Adapting school curricula and sports programs to allow for physical activity and social connection in a safe manner may help to promote mental health among high school student-athletes. Resources that promote outdoor physical activity in cooler temperatures may help to foster a sense of normality and the continued engagement in physical activity, especially in colder climates [51].

### Strengths and Limitations

A phenomenographical design allowed for the expression and understanding of variations of student-athletes’ experiences throughout the COVID-19 pandemic to be captured. The wide range of perspectives illustrate the importance of individual experiences within the broader context of the pandemic, with each participant’s experience and perception valued and represented in the results. The inclusion of physical activity, mental health, and social connections helps to better understand how these health determinants intersect, and how they might shape each other in the health and well-being of student-athletes. The study findings highlight the variation in potential strategies that could be used to improve high school student-athletes’ experiences as pandemic restrictions continue, or should another pandemic occur. Participant experiences described in this study also highlight that it is not necessarily a one-size-fits-all approach, and that tailored approaches, or alternatives to the norm should be suggested as resources and supports are developed for high school student-athletes.

Although the study asked about changes in mental health throughout the COVID-19 pandemic restricts in winter and spring of 2020, we did not inquire about participants’ natural fluctuation in mental health and thus alterations in mental health reporting may have been due to external factors (e.g., seasonal changes). Additionally, to promote discussion and reduce stigma, the term “mental wellness” was used with participants rather than mental health, which may have impacted whether participants were willing to speak openly about struggles with their mental health. Further, interviews via phone may have provided less opportunity for rapport building, as body language was lost in the communication, limiting a sense of security and safety when talking about mental health. However, questions pertaining to mental health were asked after rapport building questions and questions pertaining to physical activity. Additionally, two females spoke openly about their mental health challenges and one male discussed change in his mood throughout the pandemic.

## 5. Conclusions

This novel phenomenographical study explored high school student-athletes’ experiences of physical activity, social connections, and mental health and well-being during the COVID-19 pandemic. This study highlights the interactions amongst these health determinants and provides insights into what resources and supports these student-athletes accessed and found helpful, and what areas of support were lacking. Findings provide direction on how best to support student-athletes moving forward.

## 6. Recommendations

Provide options for physical activity programs via synchronous online platforms that allow for participant engagement and observation of body language through the platformShare and disseminate free or low-cost fitness applications, websites, or YouTube channels with youth to facilitate their engagement in physical activityNo equipment or low equipment exercise programs should be disseminated to schools, sports organizations, and youthCities and communities should allocate increased walking space or green space to facilitate physical distancing when outdoorsIn colder climates, increased opportunities for cold-weather activities (e.g., skating, cross-country skiing, walking) should be facilitated by the city or community (e.g., plowing sidewalks and paths, creating cross-country ski tracks)Schools should facilitate increased opportunity to seek support from teachers (e.g., office hours where students can ask questions)Schools should implement mental health programming and wellness initiativesSchools and sports organizations should develop peer support programsMental health resources and supports and how to access the supports should be communicated to youthCommunicating information to youth should be done via social media (specifically Instagram), through teachers and school boards, and to parents and coaches.

## Figures and Tables

**Figure 1 ijerph-18-03515-f001:**
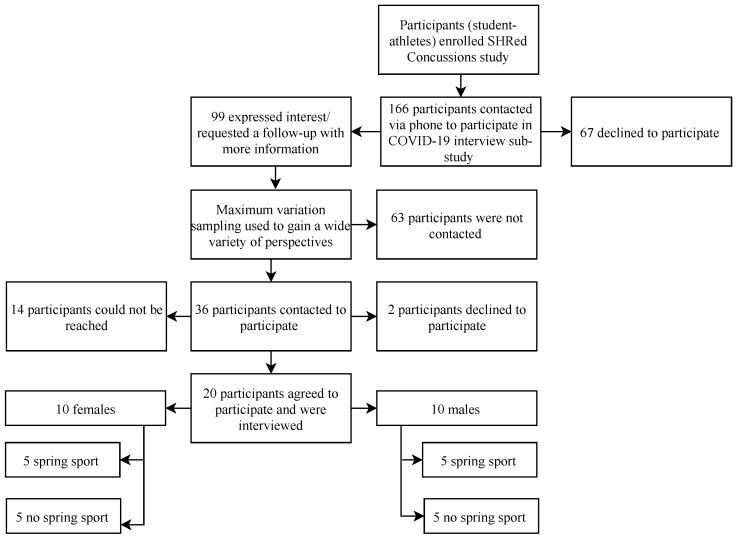
Overview of participant recruitment and data collection.

**Figure 2 ijerph-18-03515-f002:**
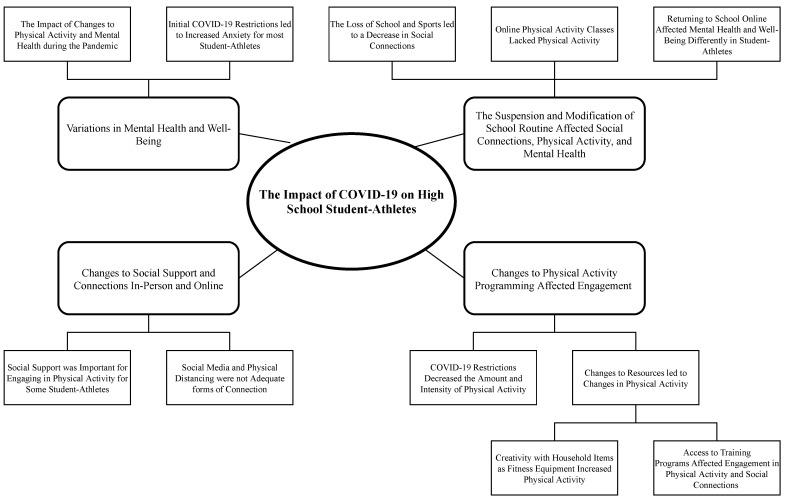
The impact of the COVID-19 pandemic on high school athlete’s experiences with physical activity, social connections, and mental health.

**Table 1 ijerph-18-03515-t001:** Participant characteristics by gender/sex, age, and spring sport, and physical activity participation.

Participant	Sex and Gender (F/M)	Age	Spring Sport (Yes/No)	Physical Activity Participation
1	F	15	Yes	Basketball, ice hockey, lacrosse, ^a^ weight training
2	M	15	Yes	Badminton, cycling, ^a^ dirt biking, football, golf, ^a^ hockey, running, skateboarding, skiing, weight training
3	M	16	No	Biathlon/cadets, football
4	F	16	Yes	Soccer ^b^
5	M	17	Yes	Football, rugby, ^a^ wrestling
6	M	16	Yes	Soccer, ^a^ track and field,^a^ weight training
7	M	16	Yes	Cycling, ^a^ football, wrestling
8	F	16	Yes	Cross-country, rugby, ^a^ running, track and field ^a^
9	F	15	No	Badminton, football, running, soccer, weight training
10	M	16	Yes	Aerobics, baseball, basketball, cycling, ^a^ football, golf, ^a^ ice hockey, lacrosse, ^a^ squash, running, volleyball, weight training
11	F	16	No	Swimming
12	F	17	No	Ice hockey, kickboxing, skateboarding, weight training, wrestling
13	M	16	No	Football, wrestling
14	F	16	Yes	Rugby, ^a^ wrestling
15	F	17	Yes	Bowling, rugby ^a^
16	M	16	No	Basketball, weight training
17	M	16	No	Basketball, weight training
18	F	15	No	Boxing, running, skiing, volleyball, weight training
19	F	15	No	Martial arts
20	M	16	No	Ice hockey, skiing

Note: ^a^ Spring sport; ^b^ Club spring/all year sport.

## Data Availability

The data are not available due to privacy and ethical restrictions.

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
