# Peer review of "The Impact of COVID-19 on High School Student-Athlete Experiences with Physical Activity, Mental Health, and Social Connection"

_ijerph, 2021, doi:10.3390/ijerph18073515_

Round 1

Reviewer 1 Report

This study is meaningful and significant in the relevant field in that this study aims to find out and describe high- school student-atheletes’ experiences and how their physical activity, mental health, and social connections were affected by the pandemic; and also in that the findings provide directions on how best to support student-athletes moving forward particularly during the COVID-10 pandemic within the Canadian context as the authors clearly mentioned in the manuscript.

However, the followings should be revised:

1. Phenomenography: Is this different from phenomenology in qualitative methods? I suppose a phenomenography or phenomeno-graphical design should be explained more in detail for readers to be able to understand the whole concept and design of the study better. Overall, it would be better for authors to strengthen the method design part.

2. Page 3 (Data Collection)

1) HAS and TE: I guess they are authors. So I think they should be mentioned as authors, since they first appear in the manuscript.

2) I understand how the authors ensured trustworthiness of the data and the process of the study. However, I am not so sure about how the authors came up with semi-structured interviews questions (or make questions)? Are those made based on previous research or on authors’ personal experience etc…? So I am just curious about the way the authors secure trustworthiness of the semi-structured-interview questions. Thus, it would be better to explain how the questions were created.

3. Page 5:

3.2.1. Part: Students’ direct responses should be italicized just like the others in the manuscript.

4. Please check stylistic errors throughout the manuscript.

Thank you.

Author Response

March 19, 2021

Dr. Paul B. Tchounwou

Editor-in-Chief

Re: The impact of COVID-19 on high school student-athlete experiences with physical activity, mental health, and social connection

Dear Dr. Tchounwou,

Thank you for considering the submission of our manuscript entitled The impact of COVID-19 on high school student-athlete experiences with physical activity, mental health, and social connection for publication in the International Journal of Environmental Research and Public Health.

Please see below our responses to Reviewer 1. We have attached a copy of the manuscript and supplementary materials with changes in tracking, and have attached clean versions of the manuscript and supplementary materials to this submission.

Reviewer 1

Comment 1: Phenomenography: Is this different from phenomenology in qualitative methods? I suppose a phenomenography or phenomeno-graphical design should be explained more in detail for readers to be able to understand the whole concept and design of the study better. Overall, it would be better for authors to strengthen the method design part.

Comment 1 Response: Although phenomenology and phenomenography share some methodological principles, phenomenography differs from phenomenology in that it employs a second-order perspective and aims to describe the differences or variations in how people or groups understand or experience a phenomenon and to categorize these descriptions.

We have added the following to the methods section to increase clarity:

Line 116-125: This study employed a second-order phenomenological perspective (phenomenography). This study employed a second-order phenomenological perspective (phenomenography). Phenomenography seeks to understand variation in the subjective nature of experiences, perceptions, and conceptualizations of a phenomenon. It aims to characterize descriptive categories that differ from one another, and can be used to identify health-related concepts that can be targeted for change.

Comment 2: HAS and TE: I guess they are authors. So I think they should be mentioned as authors, since they first appear in the manuscript.

Comment 2 Response: Thank you for your feedback. We have included additional information here.

Please see Line 147-148. All interviews were completed by HAS and TE (first and second author, respectively) in June 2020, during Phase 1 when certain COVID-19 restrictions (e.g., day camps, summer school, parks) were lifted in Calgary, Alberta

Comment 3: I understand how the authors ensured trustworthiness of the data and the process of the study. However, I am not so sure about how the authors came up with semi-structured interviews questions (or make questions)? Are those made based on previous research or on authors’ personal experience etc…? So I am just curious about the way the authors secure trustworthiness of the semi-structured-interview questions. Thus, it would be better to explain how the questions were created.

Comment 3 Response: Questions were designed to be in alignment with the phenomenographical methodology. As there was little existing research exploring COVID-19 at the time of the interviews, the questions were developed by the author team through an iterative process with experts in the fields of mental health, physical health, and social relationships and then piloted with university students and high school students.

We have added the following to the Interview Guide in Supplementary 1:

Interview Guide Development

The interview guide was developed in alignment with the phenomenography methodology. Questions within the guide were developed through an iterative process with experts in the fields of qualitative research (MHM, PDB, KLB, AMB) and with research and clinical experts in fields of physical activity, social connections, and mental health (MHM, PDB, KLB, AMB, DCKS, KOW, HAS). All authors contributed to the development of the interview guide. After an initial development phase, the interview guides were piloted with university and high school students and adjustments made to the questions, as necessary.

Comment 4: 3.2.1. Part: Students’ direct responses should be italicized just like the others in the manuscript.

Comment 4 Response: Thank you for pointing out this stylistic error. It has been corrected in the manuscript.

Please see Line 221-225: Most participants reported their mental health worsened at the onset of COVID-19 restrictions, expressing feelings of anxiousness, fear, and shock. Participant 15-F shared, “I had anxiety attacks all throughout the first week”. Two participants reported no change in their mental health, and one reported the cancellation of school “felt like…a longer and extended weekend” (P6-M).

Comment 5: Please check stylistic errors throughout the manuscript

Comment 5 Response: We believe all stylistic errors have been resolved throughout the manuscript

Thank you to the reviewers for their comments and suggestions. We hope our modifications to the manuscript and comments provided sate the review and editorial team.

Thank you for your consideration.

Sincerely,

Heather Shepherd, Amanda Black, and the authorship team.

Heather Shepherd, MScOT

Occupational Therapist

Doctoral Candidate

Sport Injury Prevention Research Centre

Faculty of Kinesiology

University of Calgary

Amanda Black, CAT(C), PhD

Certified Athletic Therapist

Assistant Professor

Sport Injury Prevention Research Centre

Faculty of Kinesiology

University of Calgary

Reviewer 2 Report

Dear author

It is a great honor for you to bring this meaningful research result. This will make a good contribution to how to take preventive measures for young athletes in the future when they face major external environmental impacts. However, please clarify and adjust the following issues.

1 Introduction

What the manuscript emphasizes is that previous studies did not investigate information about athletes, and the use of qualitative methods as an investigation method did not even appear. However, there are still existing research results for the scope of the investigation of teenagers or high school students, supplementing some existing research literature to enrich the research background of the manuscript, and emphasizing the importance of research on young athletes, which should be complete.

For example:

Aditya Thakur (2020). Mental Health in High School Students at the Time of COVID-19: A Student's Perspective. J Am Acad Child Adolesc Psychiatry, 59(12), 1309–1310. doi: 10.1016/j.jaac. 2020.08.005

  1. Method

Research methods are very important methods. However, although the methods used by the researchers are explained in the manuscript. However, because readers have different habits, it may be easier to read if you can include a clear flow chart to illustrate.

  1. Overview and Discussion The manuscript clearly explains the research objects and sports expertise. In the research and analysis, we can also understand the psychological impact of COVID19 on the local young athletes, the personal and social barriers, the inconvenience during training, and the psychological adjustment state of the school curriculum strain on the children. influences.
  2. However, the manuscript has clear information on different genders, so does the difference between different genders make a difference?
  3. The stress resistance and exercise intensity faced by young athletes during exercise will have a certain strengthening effect on physical and mental health, and different exercise properties and environments will also have different regulatory effects.

E.g: KR Fox (1999). The influence of physical activity on mental well-being. Public health nutrition, 2(3a), 411–418.

P Salmon (2001). Effects of physical exercise on anxiety, depression, and sensitivity to stress: a unifying theory. Clinical Psychology Review, 21(1), 33-61. https://doi.org/10.1016/S0272-7358 (99)00032-X

TO Filgueira, A Castoldi, LER Santos, GJ Amorim, MSS Fernandes, WLN Anastácio, EZ Campos, TM Santos, FO Souto (2021). The Relevance of a Physical Active Lifestyle and Physical Fitness on Immune Defense: Mitigating Disease Burden, With Focus on COVID-19 Consequences. Frontiers in Immunology. https://doi.org/10.3389/fimmu.2021.587146

Y Feito, KM Heinrich, SJ Butcher, WSC Poston (2018). High-intensity functional training (HIFT): definition and research implications for improved fitness. Sports, 6(3), 76. https://doi.org/10.3390 /sports6030076 Therefore, extracting the feelings of different genders or different athletes will have more research highlights.

  1. Conclusions

Take out the research suggestions, summarize them, and present your suggested directions in a list.

This is a very meaningful and interesting topic. However, qualitative research is to capture more in-depth discussions. Therefore, if relevant suggestions can be supplemented, the depth of the manuscript will be enhanced.

wish you all the best.

Author Response

March 19, 2021

Dr. Paul B. Tchounwou

Editor-in-Chief

Re: The impact of COVID-19 on high school student-athlete experiences with physical activity, mental health, and social connection

Dear Dr. Tchounwou,

Thank you for considering the submission of our manuscript entitled The impact of COVID-19 on high school student-athlete experiences with physical activity, mental health, and social connection for publication in the International Journal of Environmental Research and Public Health.

Please see below our responses to Reviewer 2. We have attached a copy of the manuscript and supplementary materials with changes in tracking, and have attached clean versions of the manuscript and supplementary materials to this submission.

Reviewer 2

Comment 1: What the manuscript emphasizes is that previous studies did not investigate information about athletes, and the use of qualitative methods as an investigation method did not even appear. However, there are still existing research results for the scope of the investigation of teenagers or high school students, supplementing some existing research literature to enrich the research background of the manuscript, and emphasizing the importance of research on young athletes, which should be complete.

For example: Aditya Thakur (2020). Mental Health in High School Students at the Time of COVID-19: A Student's Perspective. J Am Acad Child Adolesc Psychiatry, 59(12), 1309–1310. doi: 10.1016/j.jaac. 2020.08.005

Comment 1 Response: Thank you for your comments and for sharing an example reference. We have included additional background information on a high school student-athlete population, and additional information on mental health and school. Please see the following additions to the manuscript:

Line 55-71: In the United States, a study of high-school student athletes reported elevated self-reported anxious and depressive symptoms in females when compared to males [15]. McGuine et al. [15] also reported increased self-reported anxious and depressive symptomatology in grade 12 students, when compared to lower grades. Schools may provide opportunities for mental health supports (e.g., guidance counsellor) [16], and without these supports in place, coupled with increased isolation, change in routine, and decreased opportunities for socialization, the mental health in high school students is of concern.

Line 92-95: This is in contrast to the results reported by McGuine et al. [15] who found that during the first few months of COVID-19, team sport athletes self-reported more anxious and depressive symptoms compared to individual sport athletes. As such, the effect of COVID-19 may be more difficult for team sport athletes.

Comment 2: Research methods are very important methods. However, although the methods used by the researchers are explained in the manuscript. However, because readers have different habits, it may be easier to read if you can include a clear flow chart to illustrate.

Comment 2 Response: Thank you for your suggestion. We have added a diagram (Figure 1) to display the process of participant recruitment and data collection. Please see Line 189-190.

Comment 3: Overview and Discussion The manuscript clearly explains the research objects and sports expertise. In the research and analysis, we can also understand the psychological impact of COVID19 on the local young athletes, the personal and social barriers, the inconvenience during training, and the psychological adjustment state of the school curriculum strain on the children. influences.

Comment 3 Response: Thank you very much for your comments on our manuscript.

Comment 4: However, the manuscript has clear information on different genders, so does the difference between different genders make a difference?

Comment 4 Response: Although we did sample participants on both sex and gender, all participants identified as cis-gendered. Given some literature suggests that high school-aged females may report increased mental health symptomatology, we wanted to explore the gender/sex differences with respect to physical activity, social connections, and mental health. However, although we did analyse for differences across genders, none were observed and thus differences between genders are not discussed extensively in the findings.

Please see Line 187-189 for our discussion on the lack of observed group differences.

Although sampled on sex, gender, and spring sport participation, no reported differences on experiences of physical activity, mental health, nor social connections were apparent between males and females or based on season of sport participation.

Comment 5: The stress resistance and exercise intensity faced by young athletes during exercise will have a certain strengthening effect on physical and mental health, and different exercise properties and environments will also have different regulatory effects.

E.g: KR Fox (1999). The influence of physical activity on mental well-being. Public health nutrition, 2(3a), 411–418.

P Salmon (2001). Effects of physical exercise on anxiety, depression, and sensitivity to stress: a unifying theory. Clinical Psychology Review, 21(1), 33-61. https://doi.org/10.1016/S0272-7358 (99)00032-X

TO Filgueira, A Castoldi, LER Santos, GJ Amorim, MSS Fernandes, WLN Anastácio, EZ Campos, TM Santos, FO Souto (2021). The Relevance of a Physical Active Lifestyle and Physical Fitness on Immune Defense: Mitigating Disease Burden, With Focus on COVID-19 Consequences. Frontiers in Immunology. https://doi.org/10.3389/fimmu.2021.587146

Y Feito, KM Heinrich, SJ Butcher, WSC Poston (2018). High-intensity functional training (HIFT): definition and research implications for improved fitness. Sports, 6(3), 76. https://doi.org/10.3390 /sports6030076

Therefore, extracting the feelings of different genders or different athletes will have more research highlights.

Comment 5 Response: Thank you for your feedback. Although we did explore the student-athletes’ self-reported engagement in physical activity throughout the initial period of the COVID-19 pandemic, we inquired about perceived changes in physical activity, but did not require them to provide specific numerical values of exertional effects or training loads. We have added additional comments on the potential benefits of training within a group or team environment, which may have been lost due to the restrictions of the COVID-19 pandemic, and to the health benefits that engagement in physical activity may offer high school student-athletes.

Please see the following additions to the manuscript:

Line 373-375: Training with others may also increase activity duration, intensity, or motivation as per the Köhler effect, which may further support engagement in physical activity.

Line 386-388: This further supports evidence that physical activity acts a protective factor for mental health and physical health [9,41,42].

Comment 6: Take out the research suggestions, summarize them, and present your suggested directions in a list.

Comment 6 Response: Thank you for this suggestion. We have made changes to the manuscript and included the following recommendations (Line 482-502).

Recommendations

  • Provide options for physical activity programs via synchronous online platforms that allow for participant engagement and observation of body language through the platform
  • Share and disseminate free or low-cost fitness applications, websites, or YouTube channels with youth to facilitate their engagement in physical activity
  • No equipment or low equipment exercise programs should be disseminated to schools, sports organizations, and youth
  • Cities and communities should allocate increased walking space or green space to facilitate physical distancing when outdoors
  • In colder climates, increased opportunities for cold-weather activities (e.g., skating, cross-country skiing, walking) should be facilitated by the city or community (e.g., plowing sidewalks and paths, creating cross-country ski tracks)
  • Schools should facilitate increased opportunity to seek support from teachers (e.g., office hours where students can ask questions)
  • Schools should implement mental health programming and wellness initiatives
  • Schools and sports organizations should develop peer support programs
  • Mental health resources and supports and how to access the supports should be communicated to youth
  • Communicating information to youth should be done via social media (specifically Instagram), through teachers and school boards, and to parents and coaches

Thank you to the reviewers for their comments and suggestions. We hope our modifications to the manuscript and comments provided sate the review and editorial team.

Thank you for your consideration.

Sincerely,

Heather Shepherd, Amanda Black, and the authorship team.

Heather Shepherd, MScOT

Occupational Therapist

Doctoral Candidate

Sport Injury Prevention Research Centre

Faculty of Kinesiology

University of Calgary

Amanda Black, CAT(C), PhD

Certified Athletic Therapist

Assistant Professor

Sport Injury Prevention Research Centre

Faculty of Kinesiology

University of Calgary

Round 2

Reviewer 2 Report

Dear author
Glad to receive your reply. This information is quite interesting, and the manuscript has been revised. This will be a meaningful study.
wish you all the best